# A *Phytophthora* effector recruits a host cytoplasmic transacetylase into nuclear speckles to enhance plant susceptibility

Haiyang Li[1,2], Haonan Wang[1,2], Maofeng Jing[1,2], Jinyi Zhu[1,2], Baodian Guo[1,2], Yang Wang[1,2], Yachun Lin[1,2], Han Chen[1,2], Liang Kong[1,2], Zhenchuan Ma[1,2], Yan Wang[1,2], Wenwu Ye[1,2], Suomeng Dong[1,2], Brett Tyler[3,4], Yuanchao Wang[1,2]*

[1]Department of Plant Pathology, Nanjing Agriculture University, Nanjing, China; [2]Key Laboratory of Integrated Management of Crop Diseases and Pests (Ministry of Education), Nanjing, China; [3]Center for Genome Research and Biocomputing, Oregon State University, Corvallis, United States; [4]Department of Botany and Plant Pathology, Oregon State University, Corvallis, United States

**Abstract** Oomycete pathogens secrete host cell-entering effector proteins to manipulate host immunity during infection. We previously showed that PsAvh52, an early-induced RxLR effector secreted from the soybean root rot pathogen, *Phytophthora sojae*, could suppress plant immunity. Here, we found that PsAvh52 is required for full virulence on soybean and binds to a novel soybean transacetylase, GmTAP1, in vivo and in vitro. PsAvh52 could cause GmTAP1 to relocate into the nucleus where GmTAP1 could acetylate histones H2A and H3 during early infection, thereby promoting susceptibility to *P. sojae*. In the absence of PsAvh52, GmTAP1 remained confined to the cytoplasm and did not modify plant susceptibility. These results demonstrate that GmTAP1 is a susceptibility factor that is hijacked by PsAvh52 in order to promote epigenetic modifications that enhance the susceptibility of soybean to *P. sojae* infection.
DOI: https://doi.org/10.7554/eLife.40039.001

***For correspondence:**
wangyc@njau.edu.cn

**Competing interests:** The authors declare that no competing interests exist.

## Introduction

Epigenetic modifications regulate numerous biological processes including growth and development and disease resistance (*Gómez-Díaz et al., 2012*). Epigenetic modifications, including chromatin remodeling by acetylation, methylation, and ubiquitination of histones, affect the expression levels of defense genes upon pathogen attack (*Zhu et al., 2016*). Several studies have indicated that epigenetic regulation can be triggered by plant-pathogen interactions. In *Arabidopsis*, the transcription factor WRKY70, which activates salicylic acid (SA)-regulated defense-response genes, is epigenetically controlled through the trimethylation of lysine 4 of histone H3 (H3K4me3) (*Alvarez-Venegas et al., 2007*). *Arabidopsis* histone methyltransferases SET DOMAIN GROUP8 (SDG8) and SDG25 regulate immune responses triggered by PAMPs (pep1 and flg22) or effectors by modulating global histone lysine methylation (*Lee et al., 2016*). Histone H2B monoubiquitination mediated by *Arabidopsis HISTONE MONOUBIQUITINATION1* (*HUB1*) is involved in protecting plants against necrotrophic fungi (*Dhawan et al., 2009*).

Histone acetylation is an important epigenetic modification involved in remodeling chromatin structure and regulating gene expression in eukaryotes (*Berger, 2007*). The levels of histone acetylation are regulated by histone acetyltransferases (HATs) and histone deacetylases (HDACs) (*Roth et al., 2001*). Previous studies have shown that histone acetylation regulates plant immunity. For example, HDAC19 regulates resistance-gene expression in response to jasmonic acid and ethylene signaling during pathogen infection in *Arabidopsis* (*Zhou, 2005*). In rice, HDT701, which deacetylates histone

**eLife digest** Just like animals, plants can become infected and diseased. Among the microbes that infect plants, one group tends to stand out. Named *Phytophthora* after the Greek for 'the plant destroyer', these fungus-like microbes cause diseases in many species of plant, including important food crops. These diseases are difficult to control, and as a result *Phytophthora* diseases cost the farming industry billions of dollars every year.

Effective control of *Phytophthora* diseases is likely to depend on scientists first gaining a better understanding of how these microbes infect plants. Also like animals, plants have an immune system to protect themselves from disease. Yet many disease-causing microbes make so-called effector proteins to overcome their hosts' defenses. Previously in 2015, researchers reported that one effector made by a species known as *Phytophthora sojae* could suppress the immune system of soybean plants during the early stages of an infection. But it was not clear how the effector achieved this.

Now, Li et al., who include many of the researchers involved in the 2015 study, go on to show that the same effector, known as PsAvh52, helps *P. sojae* to infect soybean plants by interacting with a previously unknown soybean enzyme. The enzyme is a transacetylase, meaning it belongs to a group of enzymes that transfer a chemical marker called an acetyl group on to other molecules including proteins.

Li et al. went on to show that the PsAvh52 effector essentially hijacks the transacetylase enzyme, moving it to a location in the cell nucleus where it could chemically modify the proteins that package the soybean plant's DNA. These chemical changes activate nearby genes that would have otherwise been switched off, and these incorrectly activated genes make the plant more susceptible to the infection.

By deciphering one of the strategies that helps *P. sojae* to infect soybean plants, Li et al. have uncovered two possible approaches that may help to get this plant disease under control. The findings highlight the effector PsAvh52 as a weapon that could be blocked; they also reveal the transacetylase enzyme as a vulnerable point in the plant that could be protected. The next step will be to explore if there are chemical or genetic means that can achieve either of these two goals.
DOI: https://doi.org/10.7554/eLife.40039.002

H4 of defense-related genes, negatively modulates plant immunity (*Ding et al., 2012*). However, the potential mechanisms by which pathogen effectors might regulate epigenetic modifications by interacting with HATs and HDACs remain unclear. Some reports have indicated that effectors secreted by plant pathogens are involved in epigenetic modifications that manipulate plant immunity. The *Phytophthora sojae* effector PsAvh23 suppresses defense gene expression to promote infection of soybean by inhibiting acetylation of lysine 9 of histone H3 (H3K9) (*Kong et al., 2017*). Geminivirus TrAP inhibits histone methyltransferase SUVH4/KYP to decrease repressive H3K9me2 and DNA methylation, resulting in activation of viral virulence and plant susceptibility genes to attenuate host defense (*Castillo-González et al., 2015*). The maize pathogen, *Cochliobolus carbonum*, produces a non-ribosomal peptide toxin, HC Toxin, that inhibits the histone deacetylase of its host, resulting in hyperacetylation of both histones and non-histone proteins (*Brosch et al., 1995*; *Ransom and Walton, 1997*; *Walley et al., 2018*). In the animal parasite, *Toxoplasma*, effectors block STAT1-dependent transcription by manipulating the histone deacetylase complex Mi-2/NuRD (*Olias et al., 2016*). Although these cases suggest that pathogen manipulation of host epigenetic processes may be a broad virulence strategy, the full spectrum of underlying mechanisms remains unexplored.

Filamentous fungi and oomycete pathogens secrete intracellular and extracellular effectors to manipulate host immunity (*Giraldo and Valent, 2013*; *Tyler et al., 2006*). Oomycete-secreted RxLR effectors are cytoplasmic effectors that possess a conserved RxLR (Arg-x-Leu-Arg) motif to facilitate translocation from pathogens into host cells (*Kale et al., 2010*; *Dou et al., 2008*). However, the various mechanisms by which RxLR effectors manipulate plant immunity remain to be fully defined. Some plant targets of RxLR effectors have been identified that regulate plant immunity. For example, the *P. infestans* effector AVR3a manipulates plant immunity by stabilizing the host E3 ligase, CMPG1 (*Bos et al., 2010*). The *P. sojae* effectors PSR1 and PSR2 suppress RNA silencing in immune signaling

pathways in plants to promote *Phytophthora* infection (*Qiao et al., 2013*). The Nudix hydrolase activity of PsAvr3b reduces host NADH and ADP-ribose levels (*Dong et al., 2011*) following activation by GmCYP1 (*Kong et al., 2015*). PsAvh262 interacts with soybean proteins that control endoplasmic reticulum stress in order to modulate plant resistance (*Jing et al., 2016*). PsAvr3c reprograms host pre-mRNA splicing to subvert plant immunity, to promote disease (*Huang et al., 2017*).

In our previous research, we found that the transcript levels of the RxLR effector gene *PsAvh52* are strongly upregulated during the early stages (0.5 – 6 h.p.i.) of *P. sojae* infection (*Wang et al., 2011*; *Ma et al., 2015*). Furthermore, PsAvh52 could suppress cell death induced by effectors and PAMPs in *Nicotiana benthamiana* (*Wang et al., 2011*) and soybean (*Ma et al., 2015*). These results indicated that PsAvh52 could potentially suppress plant defense responses during the early stages of pathogen infection. In this study, we examined whether PsAvh52 is required for pathogenesis and the mechanism by which it suppresses plant immunity. We found that PsAvh52 was required for the full virulence of *P. sojae*, and promoted *P. sojae* infection when overexpressed in soybean hairy roots. PsAvh52 interacted with GmTAP1, a novel acetyltransferase, causing GmTAP1 to relocate into the nucleus. In the nucleus, GmTAP1 acetylated core histones, up-regulating the expression of potential plant susceptibility genes. PsAvh52 targeted a different set of host genes than the effector that suppresses acetylation, PsAvh23, and at an earlier time point than PsAvh23.

## Results

### PsAvh52 is a required factor for the full virulence of *P. sojae*

Genome sequence analysis found that PsAvh52 is conserved in four primary *P. sojae* isolates (*Figure 1—figure supplement 1A*). This indicated that PsAvh52 may be important for pathogenicity. To explore the role of PsAvh52 in pathogenesis, we deleted the *PsAvh52* gene from *P. sojae* using CRISPR/Cas9 (*Fang and Tyler, 2016*). Based on genomic DNA PCR analysis, we identified two knockout transformants (T33 and T37) (*Figure 1—figure supplement 1B,C*). There was no growth difference in vitro between WT, CK and Δ*PsAvh52* transformants (*Figure 1—figure supplement 1D, E*). However, the two knockout transformants produced smaller lesions on soybean seedlings compared to WT and CK (*Figure 1A*). Furthermore, the biomass of *P. sojae* was lower in soybean seedlings infected by transformants T33 and T37 than by WT and CK (*Figure 1B*). These results indicate that *PsAvh52* is required for the full virulence of *P. sojae*.

To further test whether PsAvh52 could promote *P. sojae* infection, GFP-PsAvh52 (without a signal peptide) was transiently expressed in soybean hairy roots using *Agrobacterium rhizogenes* (K599)-mediated transformation. The soybean hairy roots were inoculated with *P. sojae* P6497 labeled by the red fluorescent protein mRFP. Compared to the EV control, expression of GFP-PsAvh52 in soybean hairy roots promoted *P. sojae* infection 48 hr post-inoculation (hpi) (*Figure 1C–E*). These data indicate that PsAvh52 is a required virulence factor that can enhance the susceptibility of soybean to *P. sojae* infection.

### PsAvh52 interacts with an acetyltransferase protein of *G. max* in vivo and in vitro

To identify host target proteins of PsAvh52, we performed co-immunoprecipitation (Co-IP) assays. We transiently expressed GFP-tagged PsAvh52 in soybean hairy roots and purified PsAvh52 and its potential plant targets using GFP affinity beads. These proteins were identified by liquid chromatography tandem-mass spectrometry (LC-MS/MS). Several proteins were identified that potentially associate with PsAvh52 (*Figure 2—source data 1*). One of these was a putative soybean transacetylase protein, which we designated as GmTAP1. Three isoforms were predicted to be encoded by the *GmTAP1* gene (*Figure 2—figure supplement 1*). Of these three, however, we could only clone the longest isoform of *GmTAP1* (Glyma.18G216900.1) from soybean cDNA. A predicted paralog of GmTAP1, GmTAP2 (Glyma.09G272400.1), shared around 85% amino acid identity according to the Phytozome genome database (*Figure 2—figure supplement 1*). To examine the specific interactions of PsAvh52 with GmTAP1 and GmTAP2, we performed Co-IP experiments. Flag-PsAvh52-RFP was transiently co-expressed with GFP-GmTAP1, GFP-GmTAP2, or GFP in *N. benthamiana*, after which the proteins were incubated with GFP affinity beads. As shown by western blotting, Flag-PsAvh52-RFP was co-immunoprecipitated with GFP-GmTAP1, but not with GFP-GmTAP2 or GFP (*Figure 2A*).

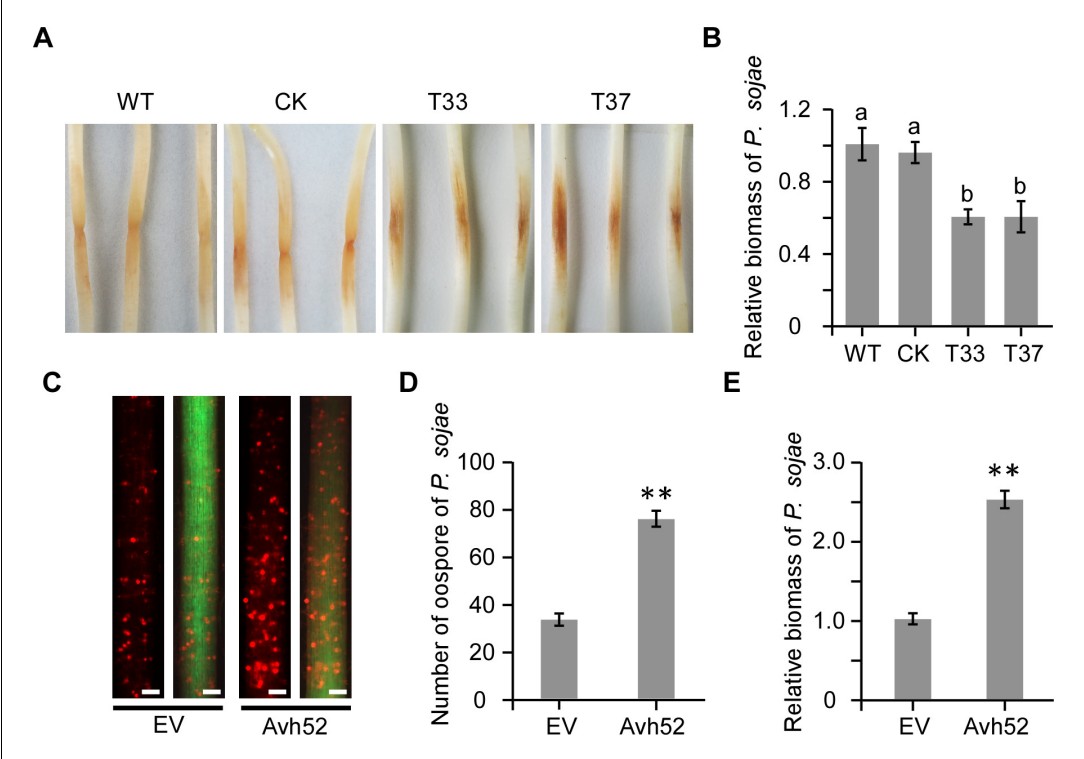

**Figure 1.** *PsAvh52* is an essential gene for full virulence of *P.sojae*. (A)-(B) Virulence of Δ*PsAvh52* knockout transformants of *P. sojae* on etiolated soybean hypocotyls. (A) Disease symptoms were observed 2 days after inoculation of the hypocotyls with 100 zoospores of *P. sojae* wild-type (P6497), knock-out mutants T33 and T37, or CK which was a non-knockout transformant recovered from the knockout transformation experiment. (B) Quantification of the biomass of *P. sojae* using genomic DNA qPCR. Data are the means ± s.d. of three independent experiments. Different letters on the graph show statistically significant differences among the samples (p < 0.01, Duncan's multiple range test) (C)-(E) Susceptibility to *P. sojae* of soybean hairy roots expressing *PsAvh52* or carrying an empty vector (EV) control. (C) Oospores produced by RFP-labeled *P. sojae* inoculated onto soybean hairy roots carrying *PsAvh52* or EV. The left image in each case shows the RFP-labeled oospores in the root, while on the right, the images of the oospores and the GFP-labeled root are combined. Scale bars = 0.2 mm. (D) Numbers of oospores of *P. sojae* were observed at 48 hpi under a confocal microscope. (E) Quantification of the biomass of *P. sojae* infection in soybean hairy roots using qPCR. In (D) and (E), data are the means ± s.d. of three independent biological replicates. Asterisks denote significant differences between samples (**p < 0.01, Student's t test).

DOI: https://doi.org/10.7554/eLife.40039.003

The following source data and figure supplement are available for figure 1:

**Source data 1.** Source data for *Figure 1*.
DOI: https://doi.org/10.7554/eLife.40039.005
**Figure supplement 1.** Knock out of *PsAvh52* using the CRISPR/Cas9 System.
DOI: https://doi.org/10.7554/eLife.40039.004

In addition, glutathione S-transferase (GST) pull-down assays were performed to confirm the interaction between PsAvh52 and GmTAP1 or GmTAP2 in vitro. The recombinant GST-GmTAP1 and GST-GmTAP2 proteins were purified from *Escherichia coli* with GST beads and incubated with His-PsAvh52. His-PsAvh52 protein was pulled down by GST-GmTAP1, but not by GST-GmTAP2 or GST alone (*Figure 2B*). To map the key region that is required for PsAvh52 interaction with GmTAP1, we constructed a series of deletion mutations and identified a mutant PsAvh52M4 (deletion of amino acids 69 – 86) that could not associate with GmTAP1 (*Figure 2C*, *Figure 2—figure supplement 2A, D,E*), demonstrating that the region of amino acids 69 – 86 is essential for the PsAvh52 interaction with GmTAP1. When PsAvh52M4 was expressed in soybean hairy roots, it could not promote *P. sojae* infection compared to PsAvh52, indicating that PsAvh52M4 lost its virulence-promoting function (*Figure 2D,E*). These results suggest that the amino acids 69 – 86 of PsAvh52 are essential for the interaction with GmTAP1 and promoting infection.

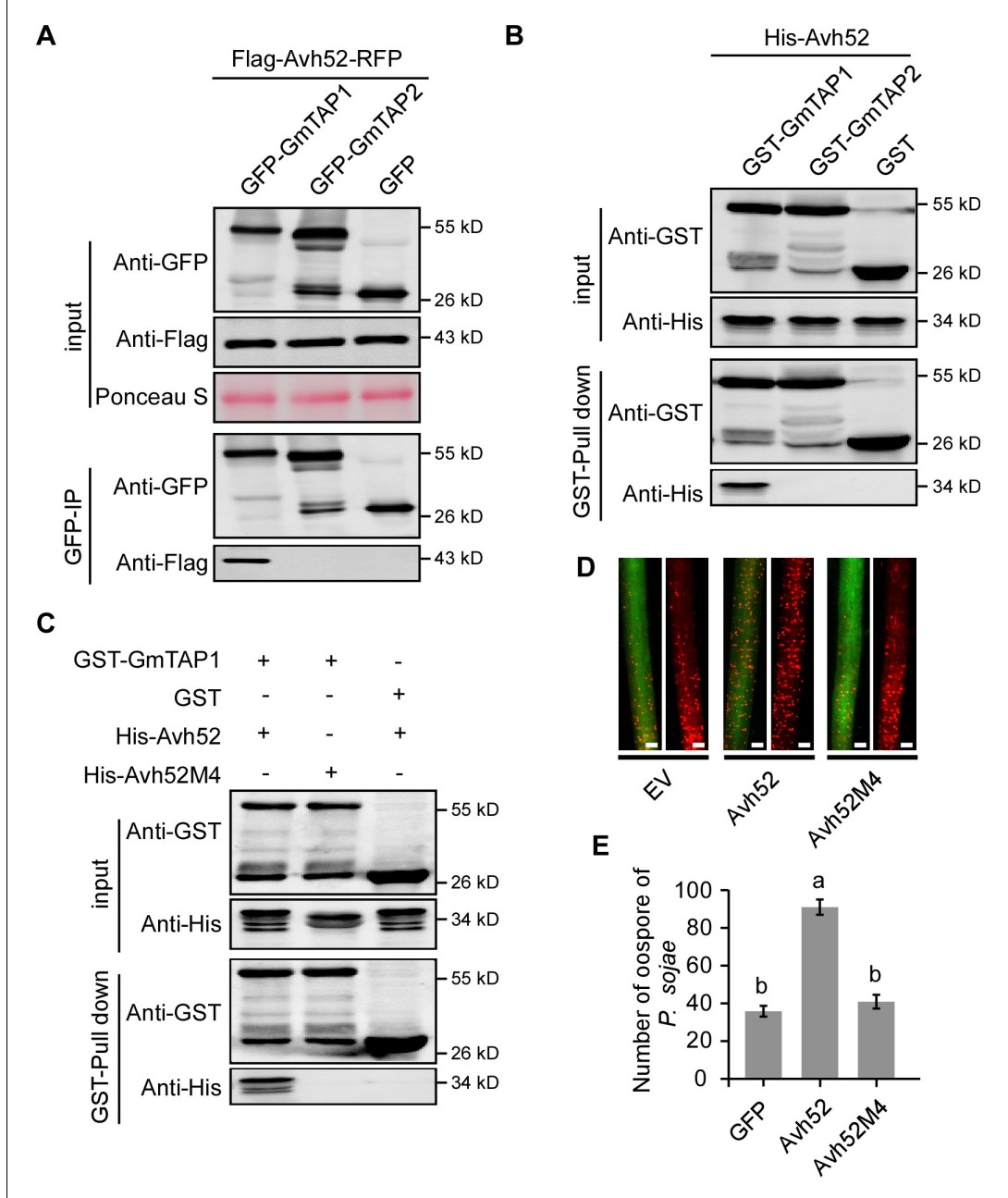

**Figure 2.** PsAvh52 binds to a soybean acetyltransferase protein, GmTAP1, in vivo and in vitro. (**A**) Co-immunoprecipitation of PsAvh52 by GmTAP1 protein, but not by GmTAP2. Total proteins were extracted from *N. benthamiana* leaves expressing Flag-Avh52-RFP with GFP-GmTAP1 or GFP-GmTAP2. Immunoprecipitation (IP) was performed using GFP-Trap beads and the captured proteins were examined by western blot using anti-Flag antibody. (**B**) In vitro pull-down of PsAvh52 by GmTAP1 but not by GmTAP2. GST-GmTAP1, GST-GmTAP2 and His-Avh52 were expressed in *E. coli* and total soluble proteins were mixed as indicated in the input. GST-pull down was performed using GST-beads and the captured proteins were examined by western blotting with anti-His antibodies. (**C**) GmTAP1 does not pull down mutant Avh52M4 in vitro. His-Avh52, His-Avh52M4 and GmTAP1 were expressed in *E. coli* and mixed as indicated. Proteins captured on GST-beads were detected by western blotting with anti-His antibodies. (**D, E**) Susceptibility to *P. sojae* of soybean hairy roots expressing PsAvh52, PsAvh52M4 or EV. Oospore production by RFP-expressing *P. sojae* was monitored at 48 hr after inoculation. Protein production was confirmed by western blots (*Figure 2—figure supplement 2C*). (**D**) Fluorescence micrographs. Scale bars = 0.25 mm (**E**) Numbers of oospores are the means ± s.d. of three independent experiments. Different letters indicate statistically significant differences among the samples (p < 0.01, Duncan's multiple range test). Similar results were observed in three independent biological experiments for (A)-(D).

DOI: https://doi.org/10.7554/eLife.40039.006

The following source data and figure supplements are available for figure 2:

*Figure 2 continued on next page*

*Figure 2 continued*

**Source data 1.** Source data for *Figure 2*.
DOI: https://doi.org/10.7554/eLife.40039.009
**Source data 2.** Source data for *Figure 2*.
DOI: https://doi.org/10.7554/eLife.40039.010
**Figure supplement 1.** Alignment of amino acid sequences of three isoforms of GmTAP1 and paralog GmTAP2 by ClustalW.
DOI: https://doi.org/10.7554/eLife.40039.007
**Figure supplement 2.** Residues 69 – 86 and the NLS (nuclear localization signal) of PsAvh52 are required for it to cause GmTAP1 to relocate into nuclear speckles.
DOI: https://doi.org/10.7554/eLife.40039.008

## PsAvh52 causes GmTAP1 to relocate into the nucleus when co-expressed *in planta*

To determine the subcellular localization of GmTAP1, we expressed GmTAP1 fused to N-terminal GFP (GFP-GmTAP1) or C-terminal RFP (GmTAP1-RFP) using *Agrobacterium*-mediated transient expression in *N. benthamiana* and observed the subcellular localization of GmTAP1 by confocal microscopy. Confocal microscopy showed that both GFP-GmTAP1 and GmTAP1-RFP localized in the cytoplasm, but not in the nucleus (**Figure 3—figure supplement 1A**). To evaluate the interaction of PsAvh52 with GmTAP1, we observed the localization of PsAvh52 and GmTAP1 during transient co-expression in *N. benthamiana*. Interestingly, when GFP-GmTAP1 and PsAvh52-RFP were co-expressed in *N. benthamiana*, GmTAP1 strongly localized in the nucleus as well as in the cytoplasm.

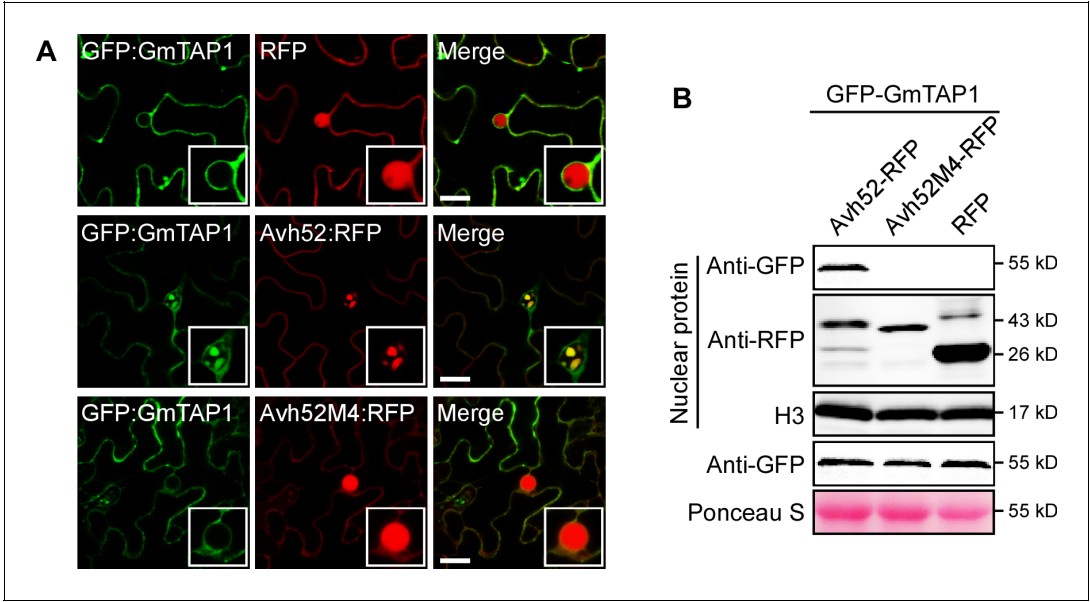

**Figure 3.** Mutant PsAvh52M4 does not cause re-localization of GmTAP1 into nuclear speckles. (**A**) The localization of GFP-GmTAP1 when co-expressed with PsAvh52-RFP, PsAvh52M4-RFP or RFP in *N. benthamiana* leaf cells. Fluorescence was detected by confocal microscopy at 24 hpi. Scale bars, 25 μm. (**B**) Western blots of nuclear and total proteins from *N. benthamiana* leaf tissue co-expressing GFP-GmTAP1 with PsAvh52-RFP fusions (WT and M4 mutant) or RFP. Proteins were detected with the indicated antibodies. Loading controls were histone H3 for the nuclear fraction and Ponceau-stained Rubisco for total proteins.
DOI: https://doi.org/10.7554/eLife.40039.011

The following source data and figure supplements are available for figure 3:

**Source data 1.** Source data for *Figure 3*.
DOI: https://doi.org/10.7554/eLife.40039.014
**Figure supplement 1.** The localization of GmTAP1 and effectors in *N.benthamiana*.
DOI: https://doi.org/10.7554/eLife.40039.012
**Figure supplement 2.** Interaction and co-localization of the N-terminal and C-terminal domains of GmTAP1 with PsAvh52.
DOI: https://doi.org/10.7554/eLife.40039.013

In the nucleus it co-localized with PsAvh52 in speckles after 24 hr expression (*Figure 3A*) and throughout the nucleus after 36 – 48 hr expression (*Figure 3—figure supplement 1B,C*). To confirm this localization pattern, we changed the fluorescent proteins and their fusion positions on PsAvh52 and GmTAP1. When GFP-PsAvh52 and GmTAP1-RFP were co-expressed in *N. benthamiana*, green and red fluorescent signals were detected in the nucleus and cytoplasm; again the proteins were localized to nuclear speckles after 24 hr (*Figure 2—figure supplement 2A*) but throughout the nucleus after 36 – 48 hr (*Figure 3—figure supplement 1B,C*). Western blot analysis showed that GmTAP1 was present in the nuclear protein fraction only when co-expressed with PsAvh52 (*Figure 3B*). These data suggested that PsAvh52 could cause the relocation of GmTAP1 into the nucleus when the two proteins were co-expressed in *N. benthamiana*. The localization pattern of GmTAP1 during *P. sojae* infection of soybean could not be tested, however, because no soybean hairy roots accumulating GFP-GmTAP1 protein could be identified, when the GFP-GmTAP1 overexpression construct was used.

The nuclear localization signal (NLS) is an important amino acid motif for import of proteins into the cell nucleus (*Kalderon et al., 1984*). It possesses one or more positively charged lysine or arginine residues, which are important for its function (*Makkerh et al., 1996*). A potential NLS was identified in PsAvh52 using the Simple Modular Architecture Research Tool (www.moseslab.csb.utoronto.ca/NLStradamus). To confirm the function of the predicted NLS and to explore whether the NLS was important for the biological function of PsAvh52, we mutated all of the lysine residues of the NLS to alanine, generating the mutant PsAvh52M1 (*Figure 2—figure supplement 2A*). The wild-type PsAvh52 was localized to the nuclear speckles, but mutant PsAvh52M1 was not localized to the nucleus at all, confirming the function of the predicted NLS (*Figure 2—figure supplement 2F*). Then we observed the localization of GmTAP1-RFP when co-expressed with GFP-PsAvh52M1 in *N. benthamiana*. In this case, neither GmTAP1 nor PsAvh52M1 were localized in the nucleus (*Figure 2—figure supplement 2A*). These results indicate that the nuclear localization of PsAvh52 was required for relocation of GmTAP1 into the nucleus. Compared to PsAvh52, the susceptibility-promoting activity of PsAvh52M1 in soybean hairy roots was no higher than the GFP control (*Figure 2—figure supplement 2B*). These results show that the nuclear localization of PsAvh52 is required for its virulence contribution and for its ability to cause GmTAP1 to relocate into the nucleus.

As PsAvh52 is a nucleus-localized effector with an NLS, we explored whether other effectors with an NLS (PsAvh53, PsAvh137) could cause GmTAP1 to relocate into the nucleus. GmTAP1-RFP and GFP-effector fusions were co-expressed in *N. benthamiana*, but red fluorescent signal was never detected in the nucleus, indicating that these other effectors with an NLS could not affect the localization of GmTAP1 (*Figure 3—figure supplement 1D–F*). These data suggested that PsAvh52 causes GmTAP1 to relocate into the nucleus through a specific interaction.

To test if direct interaction between GmTAP1 and PsAvh52 was required for the relocation of GmTAP1 into the nucleus, the GmTAP1-non-interacting mutant PsAvh52M4 was tested for its ability to cause GmTAP1 to relocate. As shown in *Figure 3*, PsAvh52M4 could not cause GmTAP1 to relocate into the nucleus when the two were transiently co-expressed in *N. benthamiana*. These results suggest that the amino acids 69 – 86 of PsAvh52, that are essential for the interaction, are also required for its ability to trigger the relocation of GmTAP1 into the nucleus. Interestingly, deletion of the interacting region caused PsAvh52 to no longer localize to the nuclear speckles; instead it remained in the nucleoplasm and the nucleolus (*Figure 2—figure supplement 2F*).

## PsAvh52 interacts with the N-terminal domain of GmTAP1

To determine which region of GmTAP1 was targeted by PsAvh52, we split GmTAP1 into two parts: the N-terminal region, GmTAP1$^{1-140}$, and the C-terminal region, GmTAP1$^{atd}$, containing the predicted acetyltransferase domain (ATD). We expressed GFP-GmTAP1, GFP-GmTAP1$^{1-140}$, GFP-GmTAP1$^{atd}$, and GFP in *N. benthamiana* leaves and purified them using GFP affinity beads. We used the purified GFP-proteins in pull-down assays with His-PsAvh52 purified from *E. coli*. His-PsAvh52 co-precipitated with GFP-GmTAP1 and GFP-GmTAP1$^{1-140}$, but not with GFP-GmTAP1$^{atd}$ nor with GFP (*Figure 3—figure supplement 2A*). This result showed that the N-terminal region of GmTAP1 interacted with PsAvh52. To confirm this interaction, we observed the localization of GFP-GmTAP1$^{1-140}$ by confocal microscopy, and found that in the absence of PsAvh52 it localized to the cytoplasm, and was excluded from the nucleus, similar to GFP-GmTAP1 (*Figure 3—figure supplement 2B*). In contrast, GFP-GmTAP1$^{atd}$ localized not only to the cytoplasm but also strongly and uniformly to the nucleoplasm

(*Figure 3—figure supplement 2B*), suggesting that the N-terminal domain is responsible for regulating the nuclear and sub-nuclear localizations of the C-terminal acetyltransferase domain, even in the absence of PsAvh52. When PsAvh52-RFP was co-expressed with GFP-GmTAP1$^{1-140}$ or GFP-GmTAP1$^{atd}$ in *N. benthamiana*, GmTAP1$^{1–140}$ re-localized from the cytoplasm to nuclear speckles together with PsAvh52-RFP after 24 hr expression and throughout the nucleus after 36 – 48 hr expression, but the localization of GFP-GmTAP1$^{atd}$ was not affected by the presence of PsAvh52-RFP (*Figure 3—figure supplement 2B,C*). Furthermore, expression of GmTAP1$^{atd}$ in *N. benthamiana* could not promote *P. capsici* infection (*Figure 3—figure supplement 2E*). Together these results indicate that interaction of the N-terminal region of GFP-GmTAP1$^{1–140}$ with PsAvh52 was required for the relocalization of GFP-GmTAP1 into the nucleus in the presence of the effector.

## Silencing of *GmTAP* genes enhances soybean resistance to *P. sojae*

Given that PsAvh52 interacted with and caused the relocalization of GmTAP1 into the nuclear speckles, we directly examined the role of GmTAP1 in plant immunity. To this end, we silenced the *GmTAP1* gene in soybean hairy roots using a hairpin RNAi construct (*Xiong et al., 2014*). Because the gene sequences of *GmTAP1* and *GmTAP2* are conserved, it was difficult to silence *GmTAP1* alone. We designed primers specific to the 3' UTR region in an attempt to specifically silence the *GmTAP1* gene, but the two *GmTAP* genes were still both silenced. The silencing constructs were transformed into soybean hairy roots via agroinfiltration. Compared to EV-transformed hairy roots, the transcript levels of the two *GmTAP* genes were significantly decreased in the *GmTAP*-silenced hairy roots (*Figure 4B*).To explore the effects of *GmTAP*-silencing on soybean resistance to *P. sojae*, *GmTAP*-silenced hairy roots were challenged with *P. sojae* labeled with mRFP. The number of oospores produced was lower in *GmTAP*-silenced hairy roots than in hairy roots carrying EV (*Figure 4A,C*).To confirm these results, we used quantitative real time PCR (qPCR) to determine the biomass of *P. sojae* in the soybean hairy roots. When *GmTAP*-silenced hairy roots were inoculated with WT *P. sojae*, the *P. sojae* biomass was only 40% of that when WT was inoculated onto control EV-carrying hairy roots (*Figure 4D*).

To confirm that GmTAP1 acts as a true virulence target of PsAvh52, the *GmTAP*-silenced hairy roots were challenged with the *P. sojae PsAvh52* deletion mutant, T37. When *GmTAP*-silenced hairy roots were inoculated with T37, the *P. sojae* biomass was no less than when T37 was inoculated onto control EV-carrying hairy roots (*Figure 4D*).These results show that the ability of *GmTAP1* to promote susceptibility to *P. sojae* depends on the expression of *PsAvh52* in the pathogen. The *P. sojae* biomass of T37 infecting *GmTAP*-silenced hairy roots was nearly the same as WT infection of the same roots (*Figure 4D*), indicating that the virulence-promoting activity of PsAvh52 is primarily dependent on the expression of GmTAP1. It is not clear if the small but statistically significant difference between T37 and WT is due to incomplete *GmTAP* silencing, additional functions of PsAvh52, other strain differences, or is not biologically relevant.

As *GmTAP2* was silenced in the RNAi experiments, we did several experiments to check the contribution of GmTAP2 to immunity. First, the localization of GmTAP2 was observed in *N. benthamiana*. We found that GmTAP2 was localized in both the plant cell nucleus and cytoplasm, which was different from the localization of GmTAP1 and resembled the localization of GmTAP1$^{atd}$ (*Figure 3—figure supplement 2B*, *Figure 4—figure supplement 1A*). As expected (since PsAvh52 and GmTAP2 do not interact physically), when GFP-GmTAP2 and PsAvh52-RFP were co-expressed in *N. benthamiana*, GmTAP2 localization was not affected by the presence of PsAvh52 (*Figure 4—figure supplement 1B*). Furthermore, expression of GmTAP2 in *N. benthamiana* could not promote *P. capsici* infection (*Figure 4—figure supplement 1C–E*), unlike GmTAP1 (see next section). These results suggested that *GmTAP2* probably did not contribute to plant immunity.

## GmTAP1 relocation to the nucleus is sufficient to promote *Phytophthora* susceptibility

In the experiments described above, we showed that PsAvh52 caused the relocation of GmTAP1 into nuclear speckles, and that the relocation of GmTAP1 into the nuclear speckles was associated with promotion of susceptibility to *P. sojae*. There could be two hypotheses regarding GmTAP1 relocalization into the nuclear speckles: (1) GmTAP1 acts as a positive regulator of immunity when localized in the cytoplasm, and relocalization into the nuclear speckles abolishes this function; (2)

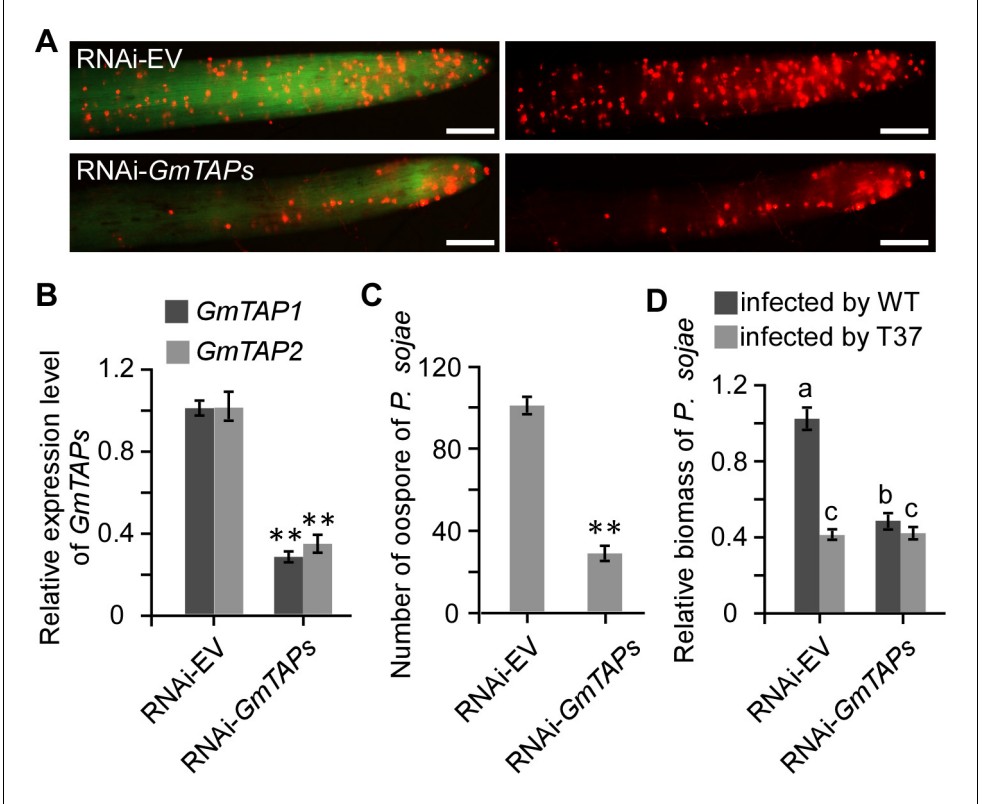

**Figure 4.** Silencing of *GmTAP* genes decreased susceptibility of soybean to *P. sojae*. Soybean hairy roots were transformed with *GmTAP*-silencing or empty vector constructs (**A**) Oospore production by RFP-labeled *P. sojae* inoculated onto *GmTAP*-silenced or EV control soybean hairy roots and monitored by fluorescence microscopy at 72 hpi. Scale bars = 0.5 mm. (**B**) Relative in dividual expression level of each *GmTAP* gene in the RNAi-*GmTAPs* hairy roots determined by qRT-PCR, relative to the levels in hairy roots carrying RNAi-EV, after normalization to the internal control, *GmCYP2*. Data are the means ± s.d. of three independent biological replicates. (**C**) Numbers of oospores produced by RFP-labeled *P. sojae* inoculated onto *GmTAP*-silenced or EV control soybean hairy roots. Numbers are the means ± s.d. of three independent biological replicates. (**D**) Virulence contribution of *PsAvh52* requires *GmTAP1* expression. The relative biomass of *P. sojae* was determined by genomic DNA qPCR 72 hr after inoculation of hairy roots with *P. sojae* wild type or Δ*PsAvh52* knockout line T37. Data are the means ± s.d. of three independent experiments. Different letters show statistically significant differences among the samples (p < 0.05, Duncan's multiple range test) In (**B**) and (**C**), asterisks denote significant differences between silenced and control roots (**p < 0.01, Student's t test).

DOI: https://doi.org/10.7554/eLife.40039.015

The following source data and figure supplement are available for figure 4:

**Source data 1.** Source data for *Figure 4*.
DOI: https://doi.org/10.7554/eLife.40039.017
**Figure supplement 1.** Localization and effect on *P.capsici* susceptibility of GmTAP2 following transient expression in *N. benthamiana* leaves.
DOI: https://doi.org/10.7554/eLife.40039.016

GmTAP1 does not affect plant immunity when localized in the cytoplasm, but acts as a negative regulator of immunity when localized in the nuclear speckles. To test these hypotheses, we inoculated *P. capsici* onto *N. benthamiana* leaves expressing various versions of *GmTAP1*. *P. capsici* does not contain any homologs of *PsAvh52*. Compared with leaves expressing the control GFP, native GmTAP1, which localized to the cytoplasm but not the nucleus (*Figure 5—figure supplement 1A*), did not affect susceptibility to *P. capsici* infection (*Figure 5A,B*). On the other hand, transient expression of GmTAP1 fused to an NLS (NLS-GmTAP1), which predominantly localized into nuclear speckles, did increase susceptibility to *P. capsici* infection. In contrast, transient expression of GmTAP1 fused to a nuclear export signal (NES-GmTAP1), which ensured that GmTAP1 remained in

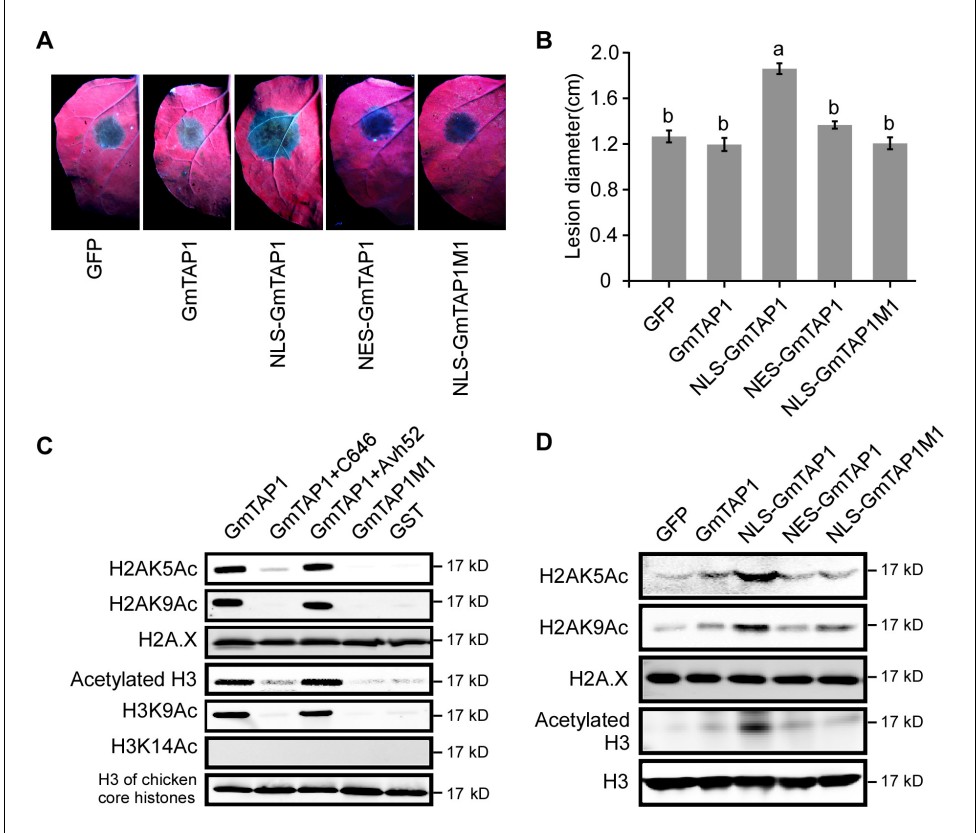

**Figure 5.** Localization to the nucleus and histone acetyltransferase activity of GmTAP1 are essential for enhancing *Phytophthora* infection. (A, B) *P. capsici* infection of *N. benthamiana* leaves expressing GmTAP1 fused to a nuclear localization signal (NLS), a nuclear exclusion signal (NES), and/or carrying an acetlytransferase-minus mutation (GmTAP1M1), compared to control leaves expressing GmTAP1 or GFP. Infection lesions were observed 48 hr after inoculation with *P. capsici* strain LT263. (A) Representative photographs under UV (B) Lesion diameters; means ±s.d. of three independent experiments. Different letters indicate statistically significant differences among the samples (p < 0.01, Duncan's multiple range test). (C, D) Histone acetyltransferase activity of GmTAP1. (C) In vitro assays using chicken core histones as substrate. Acetylation was determined by western blotting using the indicated H3ac, H2Aac, H2A.X or H3 antibodies. C646 is a chemical inhibitor of histone acetyltransferase. (D) In vivo assays following expression of GFP, GmTAP1, NLS-GmTAP1, NES-GmTAP1, or NLS-GmTAP1M1 in *N. benthamiana* leaves. After 2 days of expression, nuclear proteins were extracted and the levels of histone acetylation were detected by western blotting as in (D). All experiments in (C) and (D) were repeated three times with similar results.

DOI: https://doi.org/10.7554/eLife.40039.018

The following source data and figure supplement are available for figure 5:

**Source data 1.** Source data for *Figure 5*.
DOI: https://doi.org/10.7554/eLife.40039.020
**Figure supplement 1.** Localization and expression of various GFP-GmTAP1 fusion proteins .
DOI: https://doi.org/10.7554/eLife.40039.019

the plant cytoplasm, did not promote infection (*Figure 5A,B*; *Figure 5—figure supplement 1A,B*). These results indicated that GmTAP1 localization into the plant nucleus is sufficient to promote plant susceptibility to *Phytophthora* infection.

## Acetyltransferase activity of GmTAP1 is essential for acetylation of host histones and enhancement of *Phytophthora* infection

Since it was predicted that GmTAP1 possesses GNAT family N-acetyltransferase enzyme activity (*Smith et al., 1998*; *Dutnall et al., 1998*), we tested whether GmTAP1 could acetylate core histones in vitro. First, GST-GmTAP1 was expressed and purified in *E. coli* (*Figure 5—figure supplement 1C*). Using chicken core histones as substrates, we found that GmTAP1 had histone acetyltransferase

(HAT) activity. GmTAP1 could acetylate lysine 9 of histone H3 and lysines 5 and 9 of histone H2A (*Figure 5C*). In addition, C646 (a HAT inhibitor (*Bowers et al., 2010*)) could inhibit the enzyme activity of GmTAP1. We further tested whether PsAvh52 affected the HAT activity of GmTAP1. His-PsAvh52 protein was purified from *E. coli* and incubated with GST-GmTAP1, after which we measured the enzyme activity. We found that the HAT activity of GmTAP1 was not affected after incubation with His-PsAvh52 (*Figure 5C*). Thus, PsAvh52 did not affect the enzyme activity of GmTAP1, consistent with its interaction with the N-terminal non-enzymatic region of GmTAP1 (*Figure 3—figure supplement 2A*). There are five key amino acid residues (167-169, 179-180) predicted to be required for HAT activity in GmTAP1. We mutated those amino acid residues to alanine and confirmed experimentally that this mutant, GmTAP1-M1, had lost HAT activity (*Figure 5C*).

Next, we explored the relationship between GmTAP1's ability to increase plant susceptibility and its enzyme activity. We transiently expressed NLS-GmTAP1-M1 or NLS-GmTAP1 in *N. benthamiana* leaves then compared *P. capsici* infection of leaves expressing the two constructs. As shown in *Figure 5A*, the mutant failed to increase plant susceptibility to *P. capsici* as compared to NLS-GmTAP1 (*Figure 5A,B*). We confirmed the localization of NLS-GmTAP1-M1 into nuclear speckles, similar to NLS-GmTAP1 (*Figure 5—figure supplement 1A*). These results confirmed that the enzyme activity of GmTAP1 was required for its contribution to susceptibility.

In order to test the histone transacetylase activity of GmTAP1 *in planta*, we transiently expressed GFP, GmTAP1, NLS-GmTAP1, NES-GmTAP1, and NLS-GmTAP1-M1 in *N. benthamiana* leaves. The nuclear proteins were then extracted and the global histone acetylation levels were evaluated. As shown in *Figure 5D*, NLS-GmTAP1 increased the acetylation levels of histones H2A (lysines 5 and 9) and H3. In contrast, GmTAP1 lacking an NLS, GmTAP1 carrying an NES, and NLS-GmTAP1M1 did not increase acetylation levels of those histones (*Figure 5D*). Furthermore, expression of GmTAP[atd] in *N. benthamiana* leaves also did not increase acetylation levels of those histones (*Figure 3—figure supplement 2D*). This result indicated that N-terminus of GmTAP1 is important for targeting of the C-terminal enzymatic domain into the nuclear speckles, which is required for GmTAP1 to acetylate histones.

## PsAvh52 increases the acetylation of soybean histones *in planta* in a GmTAP1-dependent manner

The experiments described in the previous section demonstrated that histone acetylation was enhanced when GmTAP1 localized to the nucleus in *N. benthamiana*. To confirm that PsAvh52-mediated relocation of GmTAP1 into the nucleus increased histone acetylation in soybean, we measured acetylation of histones H2A and H3 in soybean hairy roots expressing either PsAvh52 or the GmTAP1-non-interacting mutant, PsAvh52M4. After *P. sojae* infection for 6 hr, nuclear H2AK5 and H3K9 acetylation were increased in PsAvh52-expressing soybean hairy roots, but the expression of PsAvh52M4 did not affect the level of H2AK5 and H3K9 acetylation (*Figure 6A*). Silencing of *GmTAP1* in soybean hairy roots followed by *P. sojae* infection for 6 hr decreased H2AK5 and H3K9 acetylation relative to the EV-carrying control roots (*Figure 5—figure supplement 1D*), suggesting that GmTAP1 is required for PsAvh52 to increase histone acetylation. To probe the ability of PsAvh52 to modify global acetylation levels of H2AK5 and H3K9 during *P. sojae* infection, we measured histone acetylation during infection of soybean hypocotyls (0 – 24 hr) by WT *P. sojae* and by the PsAvh52 deletion mutant T37 (*Figure 6B*). During early-stage (0 – 9 hr) infection by WT, both H3K9 and H2AK5 acetylation increased strongly, but no increase was observed during early-stage infection by T37 (*Figure 6B*). In a previous report, we found that during *P. sojae* infection, H3K9 acetylation increased first (0 – 6 hr) and subsequently decreased (12 – 24 hr) (*Kong et al., 2017*). This result aligns with the expression profile of PsAvh52, which showed higher expression during infection at 0.5 – 6 hr, suggesting that PsAvh52 acts primarily during early infection (*Ma et al., 2015*).

H2AK5ac and H3K9ac are epigenetic modifications associated with gene transcriptional activation (*Berger, 2007*). Since PsAvh52 promotes susceptibility through GmTAP1-mediated histone acetylation, it is plausible that PsAvh52 might cause an increase in expression of susceptibility genes. To test this hypothesis, we examined the expression of soybean orthologs of several reported susceptibility genes that play a role during pathogen-host interactions (*van Schie and Takken, 2014*). Using qRT-PCR, we measured the expression levels of soybean orthologs of a SWEET gene (*MtN3*, *Glyma.08G010000*) (*Oliva and Quibod, 2017*; *Gebauer et al., 2017*), a cell wall-degrading

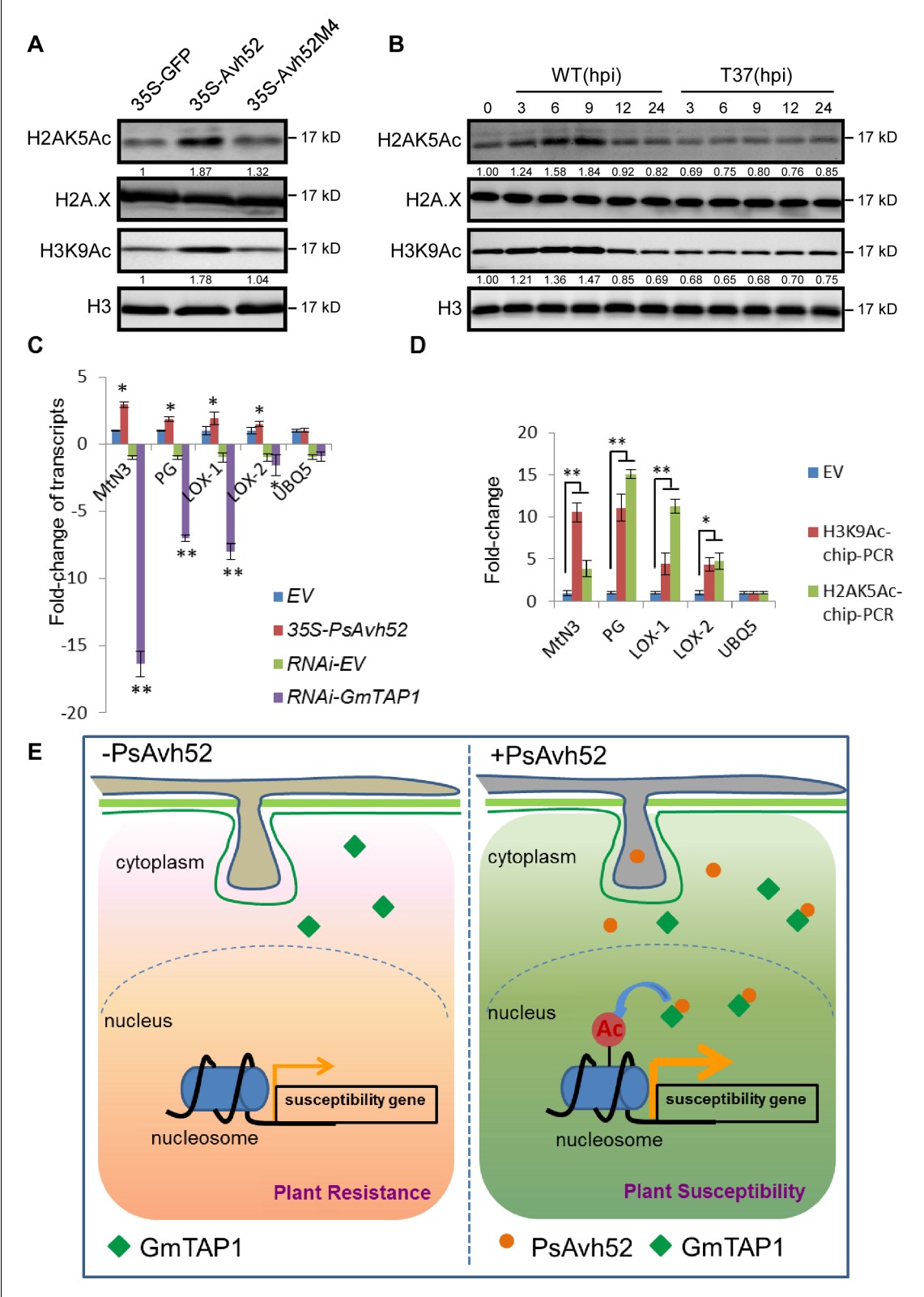

**Figure 6.** PsAvh52 increases the acetylation of soybean histones *in planta* and modulates host gene expression. (**A**) Acetylation levels of H2AK5 and H3K9 in soybean hairy roots expressing PsAvh52, PsAvh52M4, or GFP. (**B**) Acetylation levels of H2AK5 and H3K9 in soybean hypocotyl lesions infected by *P. sojae* WT or deletion mutant T37 at different time points. In (**A**) and (**B**), acetylated histones, and the controls H3 and H2A.X were detected in the nuclear protein fraction by western blotting with the corresponding antibodies. (**C**) Relative transcript levels of putative susceptibility genes in *PsAvh52-*

*Figure 6 continued on next page*

*Figure 6 continued*

expressing and *GmTAP*-silenced soybean hairy roots after infection with WT *P. sojae* strain P6497 for 6 hr. Levels were normalized to the levels in the EV overexpression control (+1) or to the RNA-EV silencing control (−1). Data are the means ± s.d. of three independent biological replicates. Asterisks denote significant differences between treatment and control (*p < 0.05 or **p < 0.01, Student's t test). (D) The histone acetylation levels of selected putative susceptibility genes were analyzed by ChIP-qPCR. Immunoprecipitation of H2AK5Ac and H3K9Ac were performed on nuclear proteins from *PsAvh52*-expressing or EV-carrying soybean hairy roots infected by *P. sojae* WT for 6 hr. Precipitation was quantitated by qPCR with specific primers, normalized to the values of the inputs, and then divided by the normalized values of the EV controls, to produce the enrichment values. Data are the means ± s.d. of three independent biological replicates. Asterisks denote significant differences between treatments and EV control (*p < 0.05 or **p < 0.01, Student's t test). (E) Working model of PsAvh52 action. PsAvh52 recruits cytoplasmically-localized GmTAP1 into nuclear speckles, where it acetylates histones H2A and H3, elevating transcription of susceptibility genes to promote *P. sojae* infection.

DOI: https://doi.org/10.7554/eLife.40039.021

The following source data and figure supplements are available for figure 6:

**Source data 1.** Source data for *Figure 6*.
DOI: https://doi.org/10.7554/eLife.40039.024
**Figure supplement 1.** Expression patterns of PsAvh52 and PsAvh23 correlate with changes in soybean histone H3 acetylation during *P.sojae* infection.
DOI: https://doi.org/10.7554/eLife.40039.022
**Figure supplement 2.** PsAvh52 and GmTAP1 do not regulate genes that are co-regulated by PsAvh23 and SAGA.
DOI: https://doi.org/10.7554/eLife.40039.023

polygalacturonase (*PG, Glyma.08G287500*) (*Nühse, 2012*; *Cantu et al., 2008*), and two lipoxygenases (*LOX-1, Glyma.13G347800, LOX-2, Glyma.08G189200*) (*van Schie and Takken, 2014*; *Gao et al., 2007*). These putative susceptibility genes were upregulated in *PsAvh52*-expressing soybean hairy roots after 6 hr of *P. sojae* infection, but downregulated in *GmTAP1*-RNAi soybean hairy roots at the same time point (*Figure 6C*). A control gene (*Ubiquitin 5; UBQ5*) showed no significant change in expression level in response to either treatment. To examine changes in histone acetylation that may be associated with gene transcriptional activation, we performed chromatin immunoprecipitation (ChIP) with H2AK5ac and H3K9ac antibody, followed by qPCR (ChIP-qPCR). Increased acetylation of histones of the four putative susceptibility genes was observed in *PsAvh52*-expressing soybean hairy roots, compared to empty vector transformed hairy roots, after 6 hr of *P. sojae* infection. In contrast, the control gene (*UBQ5*) showed no significant change in histone acetylation (*Jaskiewicz et al., 2011*) (*Figure 6D*). These results are consistent with a model in which PsAvh52 acts to increase the transcript levels of key susceptibility genes.

## Discussion

Increasing evidence suggests that epigenetic modifications including histone acetylation are an important mediator of plant defenses during plant-pathogen interactions (*Zhu et al., 2016*; *Alvarez et al., 2010*). However, the ability of plant pathogens to use effectors to disturb or co-opt host epigenetic modifications to promote infection has not been thoroughly explored. In this study, we found that the *P. sojae* effector PsAvh52, expressed in soybean hairy roots, promoted *P. sojae* infection. PsAvh52 could bind to the soybean acetyltransferase, GmTAP1, causing it to re-localize from the cytoplasm into nuclear speckles, a process that required the PsAvh52 NLS as well as protein-protein interactiondomains on both PsAvh52 and GmTAP1. Furthermore *GmTAP*-silencing in soybean hairy roots by RNAi reduced susceptibility to *P. sojae*. Finally, when GmTAP1 was localized into the nuclear speckles, it could increase the level of histone H2AK5 and H3K9 acetylation and enhanced plant susceptibility, both of which required the histone acetyltransferase enzyme activity of GmTAP1. Together, these results indicate that PsAvh52 recruits GmTAP1 to promote plant susceptibility through epigenetic modifications.

HATs include two broad classes based on their subcellular localization; namely, nuclear localization and cytoplasmic localization. The cytoplasmic HATs acetylate newly synthesized histones prior to their assembly into nucleosomes (*Lee and Workman, 2007*; *Boycheva et al., 2014*). Although we only observed cytoplasmic localization by GmTAP1 in the absence of PsAvh52, several pieces of evidence suggest that it may be targeted to the nucleus under certain conditions. First, it belongs to the GNAT family of nuclear-targeted histone acetylases (*Lee and Workman, 2007*; *Boycheva et al., 2014*). Second, it acetylated histones H2A and H3, whereas canonical cytoplasmic HATs are specific

for histone H4 (*Lee and Workman, 2007*; *Boycheva et al., 2014*). Third, when it was targeted to the nucleus by addition of an NLS, it displayed a specific sub-nuclear localization (nuclear speckles) suggesting that it had natural targets in the nucleus. Fourth, when its N-terminal PsAvh52-interaction domain was removed, it localized substantially into the nucleus, but it did not localize in the nuclear speckles nor could it promote *Phytophthora* infection. Thus, the N-terminal domain may regulate the sub-cellular localization of GmTAP1, excluding it from the nucleus as the default state, but targeting it to the nuclear speckles when it is transported into the nucleus. Together these results suggest that, under certain physiological conditions and/or in certain tissues, GmTAP1 may relocate into the nuclear speckles to modify histone acetylation, and that relocation into the nucleus may be regulated by proteins that interact with the N-terminal domain. If this is true, then PsAvh52 has evolved to subvert the normal process by which the activity of GmTAP1 activity is regulated. PsAvh52 localized to the nuclear speckles and this localization required the M4 GmTAP1-interaction domain. However, since *N. benthamiana* contains a GmTAP1 homolog, our data do not reveal whether PsAvh52 has an intrinsic speckle-targeting capability, or whether it relies on binding to GmTAP1 to target the speckles. Our current model is that PsAvh52 physically recruits GmTAP1 into the nucleus as a protein-protein complex, and that once in the nucleus, the complex is targeted to nuclear speckles through speckle-targeting signals on GmTAP1 at least, and perhaps on PsAvh52 also. However, since we did not identify a protein complex containing nuclear importins, PsAvh52 and GmTAP1 (and did not try to), we cannot rule out other mechanisms that do not involve PsAvh52 and GmTAP1 entering the nuclear as a physical complex. Nuclear speckles have most often been described as complexes involved in pre-mRNA processing (*Galganski et al., 2017*). Our data do not indicate whether the speckles targeted by PsAvh52 and GmTAP1 are also involved in pre-mRNA processing or whether we have observed a new class of speckles.

We note that extended over-expression of PsAvh52 and GmTAP1 for 36 hr to 48 hr in *N. benthamiana* cells resulted in an apparent uniform distribution of the two proteins throughout the nucleus rather than concentration in the nuclear speckles. Our data do not distinguish whether the nuclear speckles had become saturated with GmTAP1-PsAvh52 resulting in the excess complex accumulating in the nucleoplasm, or whether the complexes dissociated from the speckles over time. Many nuclear speckles are dynamic structures and proteins associated with nuclear speckles may diffusely distribute throughout the nucleoplasm (*Sleeman et al., 1998*).

GmTAP2 showed 85% sequence identity with GmTAP1 in soybean, differing primarily in the N-terminus. When expressed in *N. benthamiana*, GmTAP2 localized to the cytoplasm and also to the nucleus, but did not localize to nuclear speckles in either the presence or absence of PsAvh52. Furthermore, GmTAP2 did not bind to PsAvh52 and expression of GmTAP2 in *N. benthamiana* did not affect plant immunity. In addition, similar to GmTAP2, expression of GmTAP1[atd] which also localized to the cytoplasm and nucleus did not affect histone acetylation or plant immunity in *N. benthamiana*. These observations are consistent with our model that in order to promote susceptibility, GmTAP1 must be targeted to certain nuclear speckles via the concerted action of the PsAvh52 and the N-terminal domain of GmTAP1.

The closest homolog of GmTAP1 in *Arabidopsis* is the nuclear shuttle protein-interacting protein (AtNSI) (*McGarry, 2003*). AtNSI was identified as binding to the nuclear shuttle protein (NSP) of cabbage leaf curl geminivirus and as having the ability to acetylate histones H2A and H3 (*McGarry, 2003*), consistent with our findings. *McGarry, 2003* did not observe that AtNSI could activate transcription, but used only an in vitro assay for this test. They proposed that AtNSI may regulate the ability of NSP to regulate the shuttling of geminivirus DNA from the nucleus to the cytoplasm (*McGarry, 2003*). Thus GmTAP1/AtNSI may have been co-opted by two very different pathogens.

GmTAP1 re-localization from cytoplasm to nucleus is a significant effect of the presence of PsAvh52. The manipulation of host protein localization by effectors is an emerging theme in pathogen virulence mechanisms. Two other *P. sojae* effectors, PsCRN63 and PsCRN115, could recruit the catalase NbCAT1 into the plant nucleus to perturb $H_2O_2$ homeostasis, and promote *Phytophthora* infection (*Zhang et al., 2015*). The *Phytophthora infestans* effector Pi03192 prevented the transport of two plant NAC transcription factors from the endoplasmic reticulum to the nucleus during infection, which promoted disease progression (*McLellan et al., 2013*). Another *P. infestans* effector, Pi04314, could bind to the host protein phosphatase one catalytic subunit (PP1c), causing it to relocalize from the nucleolus to the nucleoplasm, promoting *P. infestans* infection (*Boevink et al.,*

*2016*). The *P. infestans* effector AVRblb2, bound to plant protease C14 at the haustorial interface, preventing its secretion (*Bozkurt et al., 2011*). Our finding that PsAvh52 caused GmTAP1 to relocate into host nuclear speckles to acetylate histones and promote susceptibility adds to this growing theme.

Previous studies have shown that protein acetylation plays a key role in host-pathogen interactions (*Gómez-Díaz et al., 2012*; *Alvarez et al., 2010*; *Robert McMaster et al., 2016*). Some pathogens enhance both host histone and non-histone protein acetylation to alter host immunity by secreting effectors that have acetyltransferase enzyme activity or that modulate host acetylases or deacetylases during a susceptible interaction. For example, HC toxin secreted by the fungal pathogen *Cochliobolus carbonum* inhibits maize histone deacetylase (HD) and increases the acetylation level of H3 and H4, which may affect transcription at specific gene promoters (*Brosch et al., 1995*; *Ransom and Walton, 1997*; *Walley et al., 2018*). Here we showed that PsAvh52 could cause GmTAP1 to relocate into nuclear speckles to enhance global acetylation of H2AK5 and H3K9 during early *P. sojae* infection. Recently, *Kong et al., 2017* showed that another *P. sojae* effector, PsAvh23, attenuated the acetylation of H3K9 by binding to the ADA2 subunit of the histone acetyltransferase complex, SAGA. Examination of the timing of expression of *PsAvh52* and *PsAvh23*, and the levels of H3K9 acetylation during infection by wild type *P. sojae*, and by *PsAvh52* and *PsAvh23* knockout strains, suggests that these two effectors may coordinately manipulate H3K9 acetylation levels to optimally promote *P. sojae* infection. *PsAvh52* transcript levels peak at 1 hr post infection and then decline steadily over the next 11 to 23 hr (*Figure 6—figure supplement 1A*; *Ma et al., 2015*). On the other hand, transcript levels of *PsAvh23* are initially very low and do not rise substantially until after 6 hr (*Figure 6—figure supplement 1A*; *Wang et al., 2011*). During the period from 6 to 12 hr, the transcript levels of *PsAvh52* and *PsAvh23* cross over (*Figure 6—figure supplement 1A*). During wild type *P. sojae* infection, the global acetylation levels of H2AK5 and H3K9 first increase (0 – 9 hr) and then sharply fall (after 12 hr). However, during infection by *P. sojae* strains carrying a *PsAvh23* deletion (and a functional *PsAvh52* gene), H3K9 acetylation level rise rapidly during infection, but never fall after 9 hr (*Figure 6—figure supplement 1B*; *Kong et al., 2017*). In contrast, during infection by a *PsAvh52* deletion strain (carrying a functional *PsAvh23* genes), H3K9 acetylation levels never rise, but remain at levels typical of late infection, and there is no further drop after 9 hr (*Figure 6B*; *Figure 6—figure supplement 1B*). Overexpression of *PsAvh23* or silencing of its target gene, *GmADA2*, strongly suppressed the expression of multiple defense genes including ones encoding transcription factors and MAP kinase kinase kinases (*Kong et al., 2017*). However, none of those genes were affected by over-expression of *PsAvh52* or by silencing of *GmTAP1* (*Figure 6—figure supplement 2*). On the other hand, several putative susceptibility genes were elevated by overexpression of *PsAvh52* or silencing of *GmTAP1* (*Figure 6*). Together, these observations suggest a model in which *PsAvh52* acts early in infection to promote H3K9 acetylation, stimulating the expression of susceptibility genes, and then *PsAvh23* acts to reverse the action of *PsAvh52*, and also to block any host-mediated increases in H3K9 acetylation that may support the expression of plant defense genes. It will be of interest to determine if there are additional mechanisms, other than timing of expression, that determine which host genes are targeted by the actions of these two effectors.

# Materials and methods

**Key resources table**

| Reagent type (species) or resource | Designation | Source or reference | Identifiers | Additional information |
|---|---|---|---|---|
| Strain (*Phytophthora sojae*) | P6497 | PMID: 1128796 | | |
| Strain (*Phytophthora capsici*) | LT263 | PMID: 28318979 | | |
| Gene (*Phytophthora sojae*) | *PsAvh52* | JGI Physo3 database | ID:356704 | |

*Continued on next page*

*Continued*

| Reagent type (species) or resource | Designation | Source or reference | Identifiers | Additional information |
|---|---|---|---|---|
| Gene (*Glycine max*) | *GmTAP1* | Phytozome database | ID: Glyma.18G216900.1 | |
| Antibody (Mouse monoclonal) | anti-GST | Abmart | M20007 | Western bloting (1:5000 dilution) |
| Antibody (Mouse monoclonal) | anti-His | Abmart | M30111 | Western bloting (1:5000 dilution) |
| Antibody (Mouse monoclonal) | anti-GFP | Abmart | M20004 | Western bloting (1:5000 dilution) |
| Antibody (Rat monoclonal) | anti-RFP | Chromotek | 5f8 | Western bloting (1:5000 dilution) |
| Antibody (Mouse monoclonal) | Anti-Flag | Sigma | F3165 | Western bloting (1:5000 dilution) |
| Antibody (Rabbit polyclonal) | anti-H3ac | Millipore | 382158 | Western bloting (1:2500 dilution) |
| Antibody (Rabbit polyclonal) | anti-H3K9ac | Millipore | 06–942 | Western bloting (1:2500 dilution) |
| Antibody (Rabbit monoclonal) | anti-H3K14ac | Abcam | ab52946 | Western bloting (1:2500 dilution) |
| Antibody (Rabbit monoclonal) | anti-H2K5ac | Abcam | ab45152 | Western bloting (1:2500 dilution) |
| Antibody (Rabbit monoclonal) | anti-H2AK9ac | Abcam | ab177312 | Western bloting (1:2500 dilution) |

## Vector construction

The *PsAvh52* gene and other *P. sojae* RxLR genes were cloned without their signal peptide coding regions from cDNA of P6497. *GmTAP* genes were cloned from cDNA of soybean cultivar Williams 82. All the constructs were cloned by homologous recombination with 15 – 20 bp of vector sequences at the 5' terminus of each primer using the ClonExpress II One Step Cloning Kit (Vazyme Biotech, C112). *PsAvh52* and *GmTAP* gene fragments were ligated into pBINGFP2 and pICH86988-RFP, which were used to express or localize proteins in plant cells. pEGX4T-2 and pET32a vectors were used to express proteins in *E. coli* for purification. PFGC5941 vector was used for gene silencing experiments in soybean hairy roots.

## Plant and microbe cultivation

*N. benthamiana* was cultivated at 24°C and soybean was cultivated at 25°C with 16 hr light and 8 hr darkness in humid conditions. *P. sojae* and *P. capsici* were cultivated at 25°C on 10% vegetable (V8) juice agar medium.

## Soybean hairy root transformation with *Agrobacterium rhizogenes* (K599)

After soybean seeds were sterilized for 4 – 6 hr in chlorine gas, they were grown in moist vermiculite. After 5 – 6 days, soybean cotyledons were collected and sterilized with 10% hypochlorous acid (HClO) for 8 min, then in 70% ethanol for 40 s, then rinsed with sterile water several times. Then a wound was cut into each cotyledon and 20 µl of a suspension of K599 cells (OD600 = 0.5) carrying the plasmid of interest was added to each wound. The inoculated soybean cotyledons were grown in 1/2 MS medium at 22°C with 16 hr light and 8 hr darkness. After 3 weeks cultivation, when soybean hairy roots had emerged, roots expressing green fluorescent protein were selected and tested for the expression of the genes of interest before inoculation with *P. sojae*.

## *Agrobacterium tumefaciens* infiltration assays in *Nicotiana benthamiana*

Recombinant vectors were introduced into *Agrobacterium tumefaciens* strain GV3101 by transformation. The transformed cells were incubated in LB medium with appropriate antibiotics at 28°C at 200

rpm for 24 – 36 hr. The bacteria were collected and washed with the infiltration buffer (10 mM MgCl$_2$, 10 mM MES, pH 5.7, and 100 µM acetosyringone) three times. After that, the bacteria were resuspended in infiltration buffer and adjusted to the required concentration before infiltration into attached leaves of 4 – 5 week-old *N. benthamiana* plants. After inoculation, the leaves were incubated at 25°C in the greenhouse for 24 hr to 72 hr depending on the purpose.

### *Phytophthora* infection assays

For *P. capsici* infection assays on agroinfiltrated *N. benthamiana* leaves, the leaves were collected 48 hr after infiltration with *A. tumefaciens*, then inoculated with 10% V8 juice agar plugs (diameter = 0.5 cm) containing freshly grown *P. capsici* LT263 mycelia. After 36 hr - 48 hr inoculation, the *P. capsici* lesions were measured and photographed under a UV lamp.

For *P. sojae* infection assays on soybean hypocotyls, mycelia were washed with sterilized water three times, after growth in 10% liquid V8 medium for 3 days, in order to produce zoospores. The zoospores were counted and adjusted to a concentration of 20,000/ml. 100 zoospores in 5 µl were inoculated onto etiolated hypocotyls of the susceptible soybean cultivar Hefeng 47, which were grown at 25°C for 4 – 5 days at dark. After two days incubation at 25°C in the dark, the hypocotyls were photographed, and root segments were collected for biomass assays.

For *P. sojae* infection assays on soybean hairy roots, soybean hairy roots expressing the constructs of interest were selected and placed on wet paper in a petri dish. The roots were inoculated with 10% V8 juice agar plugs containing freshly grown RFP-labeled *P. sojae* (P6497-mRFP). After 48 − 72 hr incubation at 25°C, the number of oospores in one microscopic field was photographed and counted under a fluorescent microscope from 20 inoculated hairy roots.

To quantitate the pathogen biomass in *P. sojae* infection sites, samples of 2 cm-length soybean hairy roots or hypocotyls from infection sites were collected and genomic DNA qPCR was used to confirm the relative biomass of *P. sojae* to soybean from a pool of 20 inoculated hairy roots or hypocotyls. The *P. sojae* actinA gene (ID: 108986) was used to evaluate the *P. sojae* biomass by comparison to soybean gene CYP2 (TC224926).

## Gene expression assays

Total RNA was extracted from soybean hairy roots using a NucleoSpin RNA II kit (Invitrogen) according to the manufacturer's instructions. To synthesize first strand cDNA, we used PrimeScript1 st Strand cDNA Synthesis Kit (Takara, D6110A) with 2 µg of RNA. Real-time qRT-PCR was performed in 20 µl reactions using Power SYBR Green (Applied Biosystems) following manufacturer's instructions. GmCYP2 was used as the internal standard gene and the gene primers were listed at *Supplementary file 1*. The reaction was conducted in an ABI Prism 7500 Fast Real-Time PCR System (Applied Biosystems Inc., Foster City, CA, USA). For each qPCR reaction, the reaction protocol was 95°C for 30 s followed by 40 cycles at 95°C for 5 s and 60°C for 34 s, followed by a dissociation step, 95°C for 15 s, 60°C for 1 min and 95°C for 15 s. Relative transcript levels were calculated using the $2^{-\Delta\Delta CT}$ method provided by the ABI 7500 System Sequence Detection Software.

## Confocal imaging of agroinfiltrated *N. benthamiana* leaves

After 24 – 48 hr following *A. tumefaciens* inoculation, localization of fluorescently labeled proteins was observed using an LSM 710 laser scanning microscope with the following excitation wavelengths: GFP, 488 nm; and RFP, 561 nm (Carl Zeiss, Jena, Germany).

## Plant protein expression and co-immunoprecipitation

*A. tumefaciens* cells carrying recombinant constructs were infiltrated into *N. benthamiana* leaves after mixing with *A. tumefaciens* cells carrying the P19 silencing suppressor in a 1:1 ratio. 48–72 hr after infiltration, leaves were harvested and ground in cold lysis buffer (10 mM Tris-Cl (pH 7.5), 100 mM NaCl, 0.5 mM EDTA, 0.5% NP-40) containing 1 mM phenylmethylsulfonyl fluoride (PMSF) and 1 × protease inhibitor cocktail (Sigma-Aldrich, St Louis, MO, USA). For immune-purification experiments, 72 hr after infiltration, the protein extracts were incubated with GFP-Trap_M beads (Chromotek, ABIN509397, Planegg-Martinsried, Germany) for 3 hr at 4°C. After that, the beads were collected and washed three times in 700 µl of washing buffer (10 mM Tris-Cl (pH 7.5), 100 mM NaCl and 0.5 mM EDTA) with 1 mM phenylmethylsulfonyl fluoride (PMSF) and 1 × protease inhibitor

cocktail. For mass spectrometry analysis of proteins, the bound proteins were eluted by adding 50 µl of 0.2 M glycine (pH = 2.5) for 30 s with constant mixing, followed by 5 µl of 1 M Tris base (pH = 10.4) for neutralization. For mass spectrometry analysis, after centrifugation at 2500 g for 2 min, the supernatants were stored at −70℃. For other purposes, the proteins were dissociated from the washed beads by boiling for 5 min at 95℃ in 20 µl washing buffer. The supernatants were then used for SDS-PAGE western blot assays.

## Protein pull-down assays

For in vitro pull-down assays, GST-tagged GmTAP proteins and His-tagged PsAvh52 protein were produced by expression of pEGX4T-2-*GmTAP1*, pEGX4T-2-*GmTAP2* and pET32a-*PsAvh52*, respectively, in *E. coli* strain Rosetta 2. The conditions for induction of protein expression were as follows: pEGX4T-2-GmTAP1, GmTAP1M1, GmTAP2 (16℃, 0.3 mM IPTG) and pET32a-PsAvh52 (16℃, 0.1 mM IPTG) for 16–20 hr. Total soluble proteins were extracted from harvested *E. coli* cells by ultrasonic cell disruption using lysis buffer (50 mM Tris, pH 8.0, 50 mM NaCl, 1 × protease inhibitor mixture, 1 mM PMSF). Pull-down assays were performed as follows. The GST-GmTAP1 or GST-GmTAP2 proteins were incubated with 50 µl glutathione agarose beads (GST beads) for 3 hr at 4℃. After that, the GST beads were washed three times with lysis buffer and incubated with total soluble proteins from *E. coli* cells expressing His-PsAvh52 for 5–10 hr at 4℃. The beads were collected and washed three times in 700 µl of washing buffer (50 mM Tris, pH 8.0, 50 mM NaCl, 1 × protease inhibitor mixture, 1 mM PMSF). The beads were then boiled for 5 min at 95℃ in 20 µl washing buffer to dissociate the proteins from beads. The co-precipitation of His-PsAvh52 was detected by western blotting using anti-His antibody.

## Western blotting

Proteins were separated in SDS-PAGE gels and the separated proteins were transferred to PVDF membranes. The membranes were then blocked using 5% non-fat milk in PBST buffer (1 × PBS + 0.1% Tween 20) (PBSTM) for 30 min at room temperature with 60 r.p.m. shaking. The appropriate antibodies were then added to the PBSTM. Antibodies used were: anti-GST (1:5000; #M20007; Abmart), anti-His (1:5000; #M30111; Abmart), Anti-Flag (1:5000; #F3165; Sigma-Aldrich), anti-GFP (1:5000; #M20004; Abmart), and anti-RFP (1:5000; #5f8; Chromotek). After addition of the antibodies, the membranes were incubated at room temperature for 2–4 hr or 4℃ for overnight; then washed three times (5 min each) with PBST buffer. The membranes were then incubated with a goat anti-mouse (Odyssey, no. 926 – 32210; Li-Cor) or anti-rabbit (Odyssey, no. 926–32211; Li-Cor) IRDye 800CW antibody (Odyssey, no. 926 – 32210; Li-Cor) at a dilution of 1:10,000 in PBSTM at room temperature for 30 min with 60 r.p.m. shaking; and followed by three washes (5 min each) with PBST. The signals were detected by excitation at 700 and 800 nm using a double color infrared laser imaging system (Odyssey, LI-COR company).

## Protein purification and HAT activity assays

For the in vitro histone acetyltransferase (HAT) assays, GST-tagged GmTAP proteins and His-tagged PsAvh52 were expressed in *E. coli* and purified on GST- or NiNTA-beads as described above, following the manufacturer's protocol.

For the HAT assay, reaction mixtures of 30 µl contained 8 µg core chicken histones (Millipore, 13 – 107), 2.5 µg Acetyl-CoA (Sigma-Aldrich, A2056), 10 µg purified GmTAP1 or GmTAP1M1 proteins in HAT assay buffer (50 mM Tris-Cl pH 8.0, 100 mM NaCl, 5 mM MgCl$_2$, 1 mM DTT, 5% glycerol). The reaction was incubated for 1 – 2 hr at 30℃. Then 7 µl 5 × SDS PAGE sample buffer was added to the mixture and heated at 95℃ for 5 min. To test whether PsAvh52 affected the enzyme activity of GmTAP1, His-PsAvh52 was incubated with GST-GmTAP1 at 4℃ for 1 hr before initiating the enzyme assay. Western blots were performed to check the levels of histone acetylation using one of the following antibodies: anti-H3ac (Millipore, 382158); anti-H3K9ac (Millipore, 06 – 942); anti-H3K14ac (Abcam, ab52946); anti-H2K5ac (Abcam, ab45152); anti-H2AK9ac (Abcam, ab177312). The levels of histone acetylation were quantified by ImageJ software.

For in vivo HAT assays, GFP, GmTAP1, NLS-GmTAP1, NES-GmTAP1 and NLS-GmTAP1M1 were expressed in *N. benthamiana* leaves as described above and then nuclear proteins were extracted as follows: (1) 6 g harvested plant tissue from each sample were ground in liquid nitrogen and then

placed into a 50 ml BD tube with 30 ml TBST (1 × TBS + 0.1% tween 20); (2) the BD tubes with samples were shaken gently at 4°C for 15 min; (3) the samples were filtered into new BD tubes with 2 layers of micro-cloth and centrifuged at 1500 g for 8 min at 4°C; (4) the supernatants were removed and each precipitate was resuspended in 800 µl 50% sucrose; (5) the suspensions were gently added under 800 µl 25% sucrose in 2 ml tubes and centrifuged at 13000 g for 1 min at 4°C; (6) steps (4) and (5) were repeated 7 times; (7) the final nuclear precipitates were resuspended in 100–300 µl nuclear lysis buffer (10 mM Tris-HCl, 1 mM EDTA, pH 8.0, 1% SDS, 1 mM phenylmethylsulfonyl fluoride, 1 × protease inhibitor cocktail) and vigorously shaken at 4°C for 30 min; (8) the lysates were centrifuged at 13000 g for 1 min at 4°C, then the supernatants were transferred into new 2 ml tubes; (9) 5 × SDS loading buffer (1/5 total volume) was added to the supernatants and the samples were boiled for 7 min. Then the samples were frozen until needed.

To test the levels of acetylation of the nuclear core histones, samples were analyzed using western blotting after 15% SDS-PAGE with the antibodies listed above.

## ChIP-PCR assays

Soybean hairy roots expressing *PsAvh52*-overexpression constructs or containing a control empty vector (EV) were inoculated with *P. sojae* and harvested 6 hr later. ChIP analysis was carried out as described (*Nagaki et al., 2003*; *Wu et al., 2011*). ChIP assays were performed after extracting chromatin. 5 g harvested plant tissue from each sample were ground in liquid nitrogen, washed twice using 10 ml M1 buffer (0.01 M Potassium Phosphate pH 7.0, 0.1 M NaCl, β-mercaptoethanol, 11.85% Hexylene glycol) then washed twice with 10 ml M2 buffer (0.01 M Potassium Phosphate pH 7.0, 0.1 M NaCl, 0.01 M MgCl$_2$, β-mercaptoethanol, 11.85% Hexylene glycol, 0.5% Triton X-100). 20 µl micrococcal nuclease (MNase: NEB M0247S) was added to digest nuclei in 2 ml MNB buffer (10% Sucrose, 0.05 M Tris-HCl, pH 7.5, 4 mM MgCl$_2$, 1 mM CaCl$_2$) for 10 min at 37°C. After then, 50 µl EDTA (0.5 M) is added to stop the nuclease digestion before adding antibodies. 3 µg antibodies (anti-H2K5ac, anti-H3K9ac) were added to the chromatin solution and agitated at 4 °C overnight. 30 µl protein A Dynabeads (Thermofisher, 10001D) were added to the chromatin-antibody solution and agitated for 6 hr or overnight at 4 °C. The beads were recovered using magnetic separation then washed successively with 1 ml buffer A (50 mM Tris-HCl, pH 7.5, 10 mM EDTA), containing 50 mM NaCl, 100 mM NaCl, or 150 mM NaCl. 400 µl elution buffer (20 mM Tris-HCl, pH 7.5, 50 mM NaCl, 5 mM EDTA, 1% SDS) was added twice to elute the DNA at 42°C for 15 mins. The two eluates were combined and extracted using phenol-chloroform isoamyl Alcohol (25:24:1) to remove remaining proteins, then the DNA was precipitated using ethanol overnight at −20°C, and washed twice with 70% ethanol. The sample was dried and dissolved in 20 µl ddH$_2$O. As controls, input DNA was recovered by phenol extraction after the nuclease digestion step. As a second control, the antibody was omitted from the immunoprecipitation procedure. To quantitate the genes present in the DNAs purified from the precipitated chromatin, the appropriate primers were added to qPCR reactions with the respective DNAs, and qPCR was performed. Specific primers (*Supplementary file 1*) corresponded to a short (~100 bp) gene-specific region within 500 bp following the transcription start site of each gene. The ratio for each gene of interest was calculated from the chromatin samples from the roots expressing the constructs of interest compared to those containing the empty vector, after normalizing to the respective inputs. Each ratio was then divided by the same ratio calculated for the negative control gene (UBQ5).

## Accession numbers

Transcript IDs of *P. sojae* genes in the JGI Physo3 database (http://genome.jgi.doe.gov/Physo3/Physo3.home.html): *PsAvh52* (ID: 356704), *PsAvh53* (ID: 287363), *PsAvh137* (ID: 288597). Soybean genes in Phytozome database (https://phytozome.jgi.doe.gov/pz/portal.html): *GmTAP1* (ID:Glyma.18G216900.1), *GmTAP2* (ID:Glyma.09G272400).

## Acknowledgements

We thank Prof. Brett Tyler (Oregon State University) for helpful suggestions and editing this paper, Prof. Wenbo Ma (University of California, Riverside) and Prof. Xiaorong Tao (Nanjing Agricultural University) for helpful suggestions. This research was supported by grants to Yuanchao Wang from China National Funds for the Science Fund for Creative Research Groups of the National Natural

Science Foundation of China (31721004), the Special Fund for Agro-scientific Research in the Public Interest (201303018), the China Agriculture Research System (CARS-004-PS14), the key programme of the National Natural Science Foundation of China (31430073).

## Additional information

### Funding

| Funder | Grant reference number | Author |
| --- | --- | --- |
| National Natural Science Foundation of China | Creative Research Groups 31721004 | Yuanchao Wang |
| Special Fund for Agro-scientific Research in the Public Interest | 201303018 | Yuanchao Wang |
| China Agriculture Research System | CARS-004-PS14 | Yuanchao Wang |
| National Natural Science Foundation of China | Key Programme 31430073 | Yuanchao Wang |

The funders had no role in study design, data collection and interpretation, or the decision to submit the work for publication.

### Author contributions

Haiyang Li, Formal analysis, Investigation, Methodology, Writing—original draft, Writing—review and editing; Haonan Wang, Maofeng Jing, Jinyi Zhu, Baodian Guo, Yang Wang, Yachun Lin, Han Chen, Formal analysis, Investigation; Liang Kong, Zhenchuan Ma, Formal analysis; Yan Wang, Suomeng Dong, Formal analysis, Writing—review and editing; Wenwu Ye, Software, Formal analysis; Brett Tyler, Writing—review and editing; Yuanchao Wang, Supervision, Funding acquisition, Writing—original draft, Project administration, Writing—review and editing

### Author ORCIDs

Haiyang Li (iD) http://orcid.org/0000-0002-8205-4875
Yan Wang (iD) http://orcid.org/0000-0001-7465-5518
Yuanchao Wang (iD) http://orcid.org/0000-0001-5803-5343

### Decision letter and Author response

Decision letter https://doi.org/10.7554/eLife.40039.028
Author response https://doi.org/10.7554/eLife.40039.029

## Additional files

### Supplementary files

• Supplementary file 1. Primers used in this study.
DOI: https://doi.org/10.7554/eLife.40039.025

• Transparent reporting form
DOI: https://doi.org/10.7554/eLife.40039.026

### Data availability

All data generated or analysed during this study are included in the manuscript and supporting files

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
