## [Decision Letter]

Thank you for submitting your article "A *Phytophthora* effector recruits a host cytoplasmic acetyltransferase to the nucleus to enhance plant susceptibility" for consideration by *eLife*. Your article has been reviewed by three peer reviewers, including Jian-Min Zhou as the Reviewing Editor and Reviewer #1, and the evaluation has been overseen by Detlef Weigel as the Senior Editor. The following individual involved in review of your submission has agreed to reveal their identity: Paul RJ Birch (Reviewer #3).

The reviewers have discussed the reviews with one another and the Reviewing Editor has drafted this decision to help you prepare a revised submission.

Summary:

In this study Wang and colleagues report a new mechanism by which *Phytophthora sojae* causes disease on soybean. They show that the RXLR effector protein *PsAvh52* is required for full virulence on soybean by targeting GmTAP1, an acetyltransferase. Convincing evidence is provided to show that GmTAP1 negatively regulates plant disease resistance through its acetyltransferase activity. The authors also show that GmTAP1 and *PsAvh52* are associated with increased acetylation of histones and upregulation of several candidate susceptibility genes in hairy roots. Importantly, GmTAP1 is localized in the cytoplasm, but translocates into nuclear speckles in the presence of *PsAvh52*. This requires both the NLS sequence and a sequence between RXLR motif and NLS in *PsAvh52*. Overall, this is nicely done work providing new mechanistic insight into pathogenesis by *Phytophthora sojae*.

Essential revisions:

1) Figure 1A/B – it would be normal and standard to complement a KO. Please complement the *avh52* KO mutant and perform virulence assays.

2) The relationship between virulence function of *PsAvh52* and nuclear speckle localization of GmTAP1 is interesting but not fully worked out. Significant amount of GmTAP1atd is localized in the nucleus, but not nuclear speckles (Figure 3—figure supplement 2B). Is overexpression of GmTAP1atd sufficient for histone acetylation and susceptibility to *P. capsici* or *P. sojae*? If not, it would further support a role of nuclear speckle localization in susceptibility.

---

## [Author Response]

Essential revisions:1) Figure 1A/B – it would be normal and standard to complement a KO. Please complement the avh52 KO mutant and perform virulence assays.

In *P. sojae* we currently only have one selectable marker in routine use, namely G418 resistance. Thus complementation of CRISPR mutants is currently challenging. In some cases it is possible to select variant transformants that have lost their G418 resistance. We attempted to do that with the *PsAvh52* KO *P. sojae* mutants. The two mutants were repeatedly passaged on vegetable (V8) juice agar medium without antibiotics (G418), but the *PsAvh52* KO *P. sojae* mutants could still grow on the vegetable (V8) juice agar medium with antibiotics. This complementation of the *PsAvh52* mutants was not feasible. We note that the same phenotypes were obtained with two independent transformants, but not with a control transformant that had not lost *PsAvh52*. We also note the extensive data presented in the remainder of the manuscript in support of the conclusion that *PsAvh52* contributed to the virulence of *P. sojae*. In particular, we point to the data of Figure 4D that shows that *PsAvh52* mutant T37 is equally virulent as WT when *GmTAP*-silenced roots are inoculated; those data support that the loss of virulence of T37 is specific and not an artifact of the transformation process.

2) The relationship between virulence function of PsAvh52 and nuclear speckle localization of GmTAP1 is interesting but not fully worked out. Significant amount of GmTAP1atd is localized in the nucleus, but not nuclear speckles (Figure 3—figure supplement 2B). Is overexpression of GmTAP1atd sufficient for histone acetylation and susceptibility to P. capsici or P. sojae? If not, it would further support a role of nuclear speckle localization in susceptibility.

In our original submission, we showed that overexpression of GmTAP1^atd^ in *N. benthamiana* could not promote *P. capsici* susceptibility. In the revised manuscript, we have added the result that overexpression of GmTAP1^atd^ in *N. benthamiana* could not increase histone acetylation. This supports the conclusion that speckle localization of GmTAP1 is required for acetylating histones and promoting susceptibility. We have modified the manuscript title accordingly.